# Long-term changes of nitrogen leaching and the contributions of terrestrial nutrient sources to lake eutrophication dynamics on the Yangtze Plain of China

**Qi Guan [1, 2, 3], Jing Tang [4, 5], Lian Feng [2], Stefan Olin [4], Guy Schurgers [3]**

[1] Taihu Laboratory for Lake Ecosystem Research, State Key Laboratory of Lake Science and Environment, Nanjing Institute of Geography and Limnology, Chinese Academy of Sciences, Nanjing 210008, China.

[2] School of Environmental Science and Engineering, Southern University of Science and Technology, Shenzhen 518055, Guangdong, China.

[3] Department of Geosciences and Natural Resource Management, University of Copenhagen, Copenhagen, Denmark.

[4] Department of Physical Geography and Ecosystem Science, Lund University, Lund, Sweden.

[5] Department of Biology, University of Copenhagen, Copenhagen, Denmark.

Correspondence: Jing Tang (jing.tang@nateko.lu.se)

**Abstract**

Over the past half-century, drastically increased chemical fertilizers have entered agricultural ecosystems to promote crop production on the Yangtze Plain, potentially enhancing agricultural nutrient sources for eutrophication in freshwater ecosystems. However, long-term trends of nitrogen dynamics in terrestrial ecosystems and their impacts on eutrophication changes in this region remain poorly studied. Using a process-based ecosystem model, we investigated the temporal and spatial patterns of nitrogen use efficiency (NUE) and nitrogen leaching on the Yangtze Plain from 1979 to 2018. The agricultural NUE for the Yangtze Plain significantly decreased from 50 % in 1979 to 25% in 2018, with the largest decline of NUE in soybean, rice and rapeseed. Simultaneously, the leached nitrogen from cropland and natural land increased with annual rates of 4.5 kg N ha$^{-1}$ yr$^{-2}$ and 0.22 kg N ha$^{-1}$ yr$^{-2}$, respectively, leading to an overall increase of nitrogen inputs to the fifty large lakes. We further examined the correlations between terrestrial nutrient sources (i.e., the leached nitrogen, total

phosphorus sources, and industrial wastewater discharge) and the satellite-observed probability of
eutrophication occurrence (PEO) at an annual scale, and showed that PEO was positively correlated
with the changes in terrestrial nutrient sources for most lakes. Agricultural nitrogen and phosphorus
sources were found to explain the PEO trends in lakes in the western and central part of the Yangtze
Plain, and industrial wastewater discharge was associated with the PEO trends in eastern lakes. Our
results revealed the importance of terrestrial nutrient sources for long-term changes in eutrophic status
over the fifty lakes of the Yangtze Plain. This calls for region-specific sustainable nutrient management
(i.e., nitrogen and phosphorus applications in agriculture and industry) to improve the water quality of
lake ecosystems.
**1 Introduction**
For the past half-century, China's demand for grain production has increased from 250 Mt in 1960 to
648 Mt in 2010 along with the growing population, industrial development, and human-diet changes
(Zhao et al., 2008; Wang and Davis, 1998). Substantial chemical fertilizers (i.e., 35 mega-tons, Mt,
nitrogen fertilizers in 2014 (Yu et al., 2019) simultaneously entered agricultural ecosystems for the
promotion of crop production. Although national grain production consequently increased from 132 Mt
in 1950 to 607 Mt in 2014 (Yu et al., 2019), such a level of fertilization has enhanced nitrogen discharge
to terrestrial and freshwater ecosystems, leading to a series of ecological and environmental concerns,
such as soil nitrogen pollution, water quality deterioration, and phytoplankton blooms (Zhang et al.,
2019; Wang et al., 2021b; Qu and Fan, 2010). It was reported that approximately 14.5 Mt N yr$^{-1}$ was
discharged to surface water ecosystems over entire of China for the period of 2010-2014, which largely
exceeded the national safe level of nitrogen discharge (i.e., 5.2 Mt N yr$^{-1}$) for the aquatic environment
(Yu et al., 2019). Such human-related nutrient enrichment poses a big challenge to China's sustainable
development goals (Wang et al., 2022).
The Yangtze Plain, with a human population of 340 million and an agricultural area of 100 million
hectares  (Chen et al., 2020b; Hou et al., 2020), is experiencing unprecedented ecological and
environmental issues (Guan et al., 2020; Feng et al., 2019). From 1990 to 2015, total crop production
increased by 15 % at the expense of an increase of 89 % in nitrogen fertilizers over the Yangtze Plain
(Xu et al., 2019). Consequently, more frequent nitrogen pollution was observed in soil and water. For
example, heavy fertilizer usage and intensive livestock contributed to soil nitrogen pollution in the
Yangtze River Delta for the past four decades, leading to soil deterioration and nitrogen discharge (Zhao
et al., 2022). Nitrogen discharge related to human activities (i.e., fertilizer and manure applications, and
human food waste) largely increased the nutrient loading and accelerated the degradation of water
quality in the Yangtze River since the 1990s (Chen et al., 2020c). Under the recent sustainable
development plans proposed by national and local governments, managing nitrogen sources from urban
and crop systems is envisaged to mitigate severe soil and water pollution (Chen et al., 2020c; Zhao et
al., 2022; Shi et al., 2020). However, for the Yangtze Plain with a variety of crops and crop management,
the lack of insights into long-term changes in nitrogen dynamics, such as fertilizer application, plant
nitrogen uptake, and nitrogen leaching, has limited our solution of proposing effective policies related
to nutrient management.
In recent several decades, national field surveys and satellite observations have been widely used to
investigate nutrient loadings (Tong et al., 2017; Li et al., 2022), chlorophyll-a concentrations (Guan et
al., 2020), trophic state index (TSI) (Hu et al., 2022; Chen et al., 2020a) and algal bloom occurrence
(Huang et al., 2020) to assess eutrophication issues in local or regional lakes of the Yangtze Plain, all
of which revealed the lakes on the Yangtze Plain experienced eutrophication and algal blooms for the
past two decades. Cyanobacteria blooms were reported to frequently occur in Taihu and Chaohu lakes,
with the peak expanded extent reported for 2006 (Qin et al., 2019). Since then, the magnitudes of algal
blooms significantly decreased from 2006 to 2013, and slightly increased again from 2013 to 2018
(Huang et al., 2020). Satellite observations revealed widespread and serious eutrophication issues in
large lakes of the Yangtze Plain for the periods of 2003-2011 and 2017-2018, although significantly
decreasing trends were found in 20 out of 50 lakes throughout the periods (Guan et al., 2020). Moreover,
35-year Landsat-derived trophic state index (TSI) also indicated that hyper-eutrophic and eutrophic
lakes mainly characterized the Yangtze Plain, with slight increase in TSI from 1986 to 2012 and then
decrease since 2012 (Hu et al., 2022). National field surveys demonstrated that although total
phosphorus concentrations overall decreased from 2006 to 2014, it still remained under high levels
(i.e., > 50 µg $L^{-1}$) in eastern China lakes (Tong et al., 2017). Various laws and guidelines were
implemented on regional and national scales to control eutrophication problems, such as the Guidelines
on Strengthening Water Environmental Protection for Critical Lakes in 2008 and the Water Pollution
Control Action Plan in 2015 (Huang et al., 2019). Nevertheless, the eutrophication issues are still
challenging to control and improve under the scarcity of effective strategies for the whole Yangtze Plain
due to the unknown causes of eutrophication issues.
To understand the primary causes of eutrophication in the lakes of the Yangtze Plain, previous studies
have attempted to determine the contributions of riverine nutrient exports and lacustrine nutrient loading
to algal blooms in individual lakes, such as Taihu and Chaohu lakes (Tong et al., 2017; Tong et al.,
2021; Xu et al., 2015). Based on field-measured phytoplankton biomass and nutrient concentrations,
algal blooms in Taihu Lake were primarily attributed to excessive nutrient loads from 1993 to 2015
(Zhang et al., 2018). Overloaded nutrients, in combination with climatic warming, were found to
regulate the seasonal variations of cyanobacteria blooms in Chaohu Lake based on the monthly nutrient
monitoring at discrete points (Tong et al., 2021). However, these studies only tracked the primary
drivers of algal blooms for individual hyper-eutrophic lakes (i.e., Taihu and Chaohu lakes), which is
insufficient to understand regional variations in terms of the causes of eutrophication and support the
design of effective management strategies to mitigate eutrophication issues across different eutrophic
states of lakes. Furthermore, lacustrine nutrient loading is always associated with terrestrial nutrient
sources, such as synthetic fertilizers, livestock manure, and industrial sewage (Wang et al., 2019b; Yu
et al., 2018). For example, Wang et al. (2019b) identified that diffuse sources contributed 90% to
riverine exports of total dissolved nitrogen, and point sources discharged 52% of riverine phosphorus
exports to Taihu Lake, where diffuse sources are synthetic fertilizers and atmospheric deposition, and
point sources are sewage and manure discharge. It was also reported that chemical fertilizer and
wastewater discharge provided primary nitrogen sources for the Chaohu Lake (Yu et al., 2018).
Unfortunately, all these studies did not examine the impacts of vegetation uptake and soil retention on
terrestrial nutrient sources, making it insufficient to comprehensively understand the linkage between
terrestrial nutrient sources and eutrophication in regional lake ecosystems.
In this study, we employed a process-based dynamic vegetation model, LPJ-GUESS (Smith et al., 2014),
to investigate terrestrial nitrogen dynamics for the past four decades, examining the primary drivers of
eutrophication trends in fifty large lakes of the Yangtze Plain (covering 63% of the whole plain). We
simulated the vegetation dynamics, nitrogen cycles for agricultural and natural ecosystems from 1979
to 2018, and then assessed the temporal trends of nitrogen use efficiency and nitrogen leaching. The
terrestrial nutrient sources were used to examine their linkage with the satellite-derived eutrophication
changes for fifty large lakes.

## 2 Materials and Methods

### 2.1 Study area

The Yangtze Plain (Fig. 1) is in the middle and lower basin of the Yangtze River. It covers a total area
of $7.8 \times 10^6$ km$^2$ from Hunan Province to Shanghai City, and accommodates approximately 5000
freshwater lakes, ponds and reservoirs (Hou et al., 2017). Its sub-tropical monsoon climate provides
annual mean temperature (~15°C) and precipitation (~1000 mm) conditions favorable for crop
cultivation, in particular cereals and oil seeds, making the Yangtze Plain one of the top three food
production regions in China. Generally, rice-sown area contributed dominantly to agriculture areas
associated with climate conditions and human diet (Piao et al., 2010; Tilman et al., 2011). To enhance
crop production, double-cropping strategy has been widely implemented on the Yangtze Plain, such as
the rotation of early- and late-season rice (Chen et al., 2017), and the rotation of summer maize and
winter wheat (Xiao et al., 2021). Several common management practices were adopted by millions of
smallholders (Cui et al., 2018). For example, straw return, organic manure applications, and suitable
planting density were also recommended in recent years (Cui et al., 2018). Significantly increased
fertilizer applications to cropland were expected to stimulate crop yield over the past half century (Yu
et al., 2019; Zhang et al., 2015). Such management practice can certainly enhance agriculture
productivity, but also cause negative consequences to soil and aquatic environment (Liu et al., 2016a;
Shi et al., 2020). However, since the policy of Reform and Opening-up of China in the 1980s (Zhang
et al., 2010), agricultural ecosystems have been confronted with great pressure from urban expansion

on the Yangtze Plain. Rapid urban expansion encroached on arable land, mainly on the eastern parts of

the Yangtze Plain (Zhang et al., 2021).

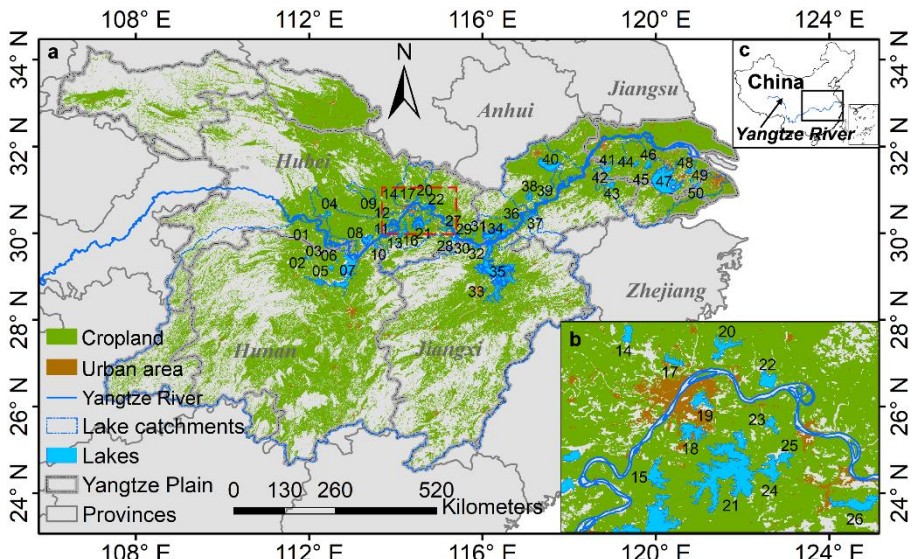

**Figure 1.** Locations of the Yangtze Plain and the fifty large lakes studied here. (**b**) Detailed overview

of Wuhan region (red box in (**a**)) and the surrounding lakes.

**2.2 Dynamic vegetation model**

We used a dynamic ecosystem model, LPJ-GUESS (Smith et al., 2014; Olin et al., 2015b), to simulate

vegetation dynamics (i.e., the establishment, growth, competition, and mortality of plants), soil

biogeochemistry, and carbon and nitrogen cycles for different ecosystems on the Yangtze Plain. The

model has been widely used to assess ecosystem carbon and nitrogen fluxes at regional and global scales

(Smith et al., 2014). Plant functional types (PFTs) and crop functional types (CFTs) are designed to

describe the different types of plants and crops with a set of pre-defined bioclimatic and physiological

parameters, such as photosynthetic pathways, phenology, growth forms and life history strategies for

PFTs, as well as irrigation, fertilization, and rotation schemes for CFTs (Smith et al., 2014; Sitch et al.,

2003; Olin et al., 2015a; Lindeskog et al., 2013).

Carbon and nitrogen fluxes between ecosystems and the atmosphere are calculated on a daily basis. For

natural PFTs, net primary production (NPP) is accumulated and allocated to different plant

compartments (i.e., leaves, roots, sapwood and heartwood for trees) at the end of each simulation year.

Soils are represented by 11 carbon and nitrogen pools with different decomposition rates (Parton et al.,
1993; Parton et al., 2010), dependent on soil temperature and texture, water content, and base decay
rates (Smith et al., 2014). Atmospheric deposition and plant biological fixation provide nitrogen sources
for plant growth and development, while the decomposition of soil organic matter can release mineral
N into the soil and nitrogen-related gases into the atmosphere. Moreover, soluble nitrogen in soil can
also leach with the surface runoff in the forms of dissolved organic and inorganic nitrogen (i.e., DON
and DIN). In the model, leaching of DON is a function of the decay rates of soil microbial carbon pool
and soil percolation, while DIN leaching depends on the available mineral nitrogen in soils and soil
percolation, as well as soil water content.
Crop growth starts from a seedling with initial carbon and nitrogen masses at a prescribed sowing date.
Chemical fertilizer and livestock manure supply external nitrogen for crop growth. According to local
farmers' practice (Shi et al., 2020), chemical fertilizer and manure applications are often applied at three
different stages: sowing, tillering, and heading stages. Such fertilization schemes are also represented
in the LPJ-GUESS (Olin et al., 2015a), where nitrogen fertilizer is applied when the crop development
stage reaches 0, 0.5, and 0.9 in response to three above stages, and the relative fertilization rate for each
stage are empirical parameters based on field surveys. Crop N uptake is simulated as the lesser between
crop N demand and accessible mineral N in soils, where the former depends on crop development stages
and C:N ratios of leaves and roots, and the latter is affected by soil temperature and fine root biomass
(Olin et al., 2015a). Differing from natural PFTs, NPP is allocated to leaves and stems, root, and storage
organs for each CFT on a daily basis, according to the daily allocation strategies related to crop
development stages (Olin et al., 2015a).

**2.3 LPJ-GUESS input, calibration, and evaluation dataset**

**2.3.1 Input data**

We ran LPJ-GUESS separating four land use types (natural land, cropland, pasture and urban) with a
500-year spin-up to simulate the vegetation dynamics and the associated nitrogen fluxes for the Yangtze
Plain from 1979 to 2018.
The gridded input data for LPJ-GUESS include climate, fractions of four land use types, total chemical
fertilizer and manure application rates, cover fractions of each CFT within the cropland area, and soil
properties. We used daily temperature, precipitation, and shortwave radiation provided by the China
Meteorological Forcing Dataset (CMFD), with a spatial resolution of 0.1° and a temporal coverage of
1979-2018 (He et al., 2020). The 300-m Climate Change Initiative Land Cover (CCI-LC version 2.0)
dataset was regrouped into four different land use types (i.e., urban, cropland, pasture, and natural land)
to obtain the cover fractions within each 0.1° grid cell for the period of 1992 to 2018 (Defourny et al.,
2012) (see the details about regrouping process in Supplementary S1). Soil properties, i.e., fractions of
sand, clay and silt, organic carbon content, C:N, pH, and bulk density were extracted from the World
Inventory of Soil Property Estimates (WISE30sec) dataset (Batjes, 2016). Based on the original data
with a spatial resolution of 30 sec, we determined the dominant FAO soil type based on their relative
area in each grid cell, and used its properties as input data for the grid cell. Gridded chemical fertilizer
and manure application data were extracted from global fertilizer usage (Lu and Tian, 2017) and manure
data (Zhang et al., 2017), which have spatial resolutions of 0.5° and 0.5', respectively. We resampled
the fertilizer and manure application data into the spatial resolution of 0.1° to represent the chemical
fertilizer and manure application for each grid cell from 1979 to 2014. The gridded monthly N
deposition data were also extracted from an external database as an input file (Lamarque et al., 2013).
It has a spatial resolution of 0.5°, and we used the value in the nearest grid cell to represent N deposition
in the simulations.
The gridded fractions of CFTs were calculated based on observational data provided by the China
Meteorological Data Service Center (https://data.cma.cn/site/subjectDetail/id/101.html). The dataset
contains the information about the types, sowing and harvest dates for a total of eleven crops at 92
observational sites across the whole Yangtze Plain (listed in Table S1). An adaptive inverse distance
weighting method was then used to interpolate the maps of the relative fractions of all crops, and their
sowing and harvest dates for the period of 1992-2015 (see the details in Supplementary S2). Due to the
limited availability for the period of 1979-1991 and 2016-2018, we used the same crop information (i.e.,
the fractions of crop types, sowing and harvest dates) from the nearest years.

**2.3.2 Model calibration and evaluation data**

The model was calibrated based on the observed crop yield collected by the China Meteorological Data Service Center (https://data.cma.cn/site/showSubject/id/102.html). The dataset provides crop yield data for eight main crops collected at different numbers of sites (i.e., winter wheat (number of sites: 37), spring maize (6), summer maize (10), single-season rice (28), early-season rice (30), late-season rice (30), rapeseed (38), and soybean (15)), for the period of 2000-2013. For the Yangtze Plain, hybrid and super-hybrid rice are widely cultivated to obtain high grain yield within short growing seasons due to the enhanced photosynthetic rates associated with leaf-level chlorophyll and rubisco contents (Huang et al., 2016). However, the default parameters for rice CFTs in LPJ-GUESS cannot capture the high-yield features of hybrid and super-hybrid rice. Therefore, we calibrated the relationship between the leaf-based nitrogen content and the maximum catalytic capacity of rubisco (see the details in Supplementary S3). We randomly selected five sites with rice yield data from 2000 to 2013 as the calibration data, and the other rice yield data were used as the evaluation data. For parameters of other CFTs (listed in Table S1), the default values performed satisfactorily in the comparison with all observed yield data (Fig. 2). It is noted that regional mean yield for each crop was derived from the evaluation data to compare the simulated values on the Yangtze Plain.

Simulated GPP and LAI were further compared with Global Solar-induced Chlorophyll Fluorescence Gross Primary Productivity (GOSIF GPP) and third generation of Global Inventory Modeling and Mapping Studies Leaf Area Index (GIMMS LAI3g) products to evaluate the performance of modelled vegetation variables. The global GOSIF GPP products have a spatial resolution of 0.05° and cover the period of 1992-2018 (Li and Xiao, 2019). Biweekly GIMMS LAI3g products with a spatial resolution of 0.25° were obtained and then converted to annual mean LAI3g maps from 1982 to 2011 (Zhu et al., 2013).

The modelled responses of nitrogen leaching to different fertilizer applications were evaluated based on an observational dataset published by Gao et al. (2016), where they collected nitrogen leaching for plots with 3 or 4 different levels of nitrogen fertilizer inputs for maize, rice, and wheat. In our study, we selected the observed responses without influences of phosphorus and potash fertilizers on the Yangtze

Plain as the evaluation data (two samples for each crop). For these sites, individual simulations were
performed by assigning the full coverage of each corresponding crop growth and prescribing the levels
of nitrogen fertilizer applications as in the experimental site. It should be noted that we used the same
nitrogen fertilizer applications in the period prior to the field experiment.
**2.4 Assessment of long-term changes in nitrogen dynamics**
We assessed long-term changes in nitrogen use efficiency (NUE) and nitrogen leaching over the past
four decades. For the LPJ-GUESS simulated NUE and leached nitrogen, a linear regression was
conducted on the annual mean values for the whole Yangtze Plain to determine the associated change
rates (i.e., the regression slopes), and the significance was tested by a *t*-test. The mean leached nitrogen
over the drainage area of all examined lakes was calculated to explore long-term changes in terrestrial
nitrogen sources for lake ecosystems, and the associated temporal trends were assessed by the linear
regression and *t*-test.
**2.5 Examination of the primary driving forces of eutrophication dynamics**
**2.5.1 Satellite-derived eutrophication changes**
We used satellite-derived PEO data published in Guan et al. (2020) to represent the eutrophication
changes for fifty large lakes on the Yangtze Plain. The PEO was defined as the frequency of high
chlorophyll-a concentrations (i.e., $> 10$ mg m$^{-3}$) or algal bloom occurrences in satellite imagery for each
year. All full-resolution (300 m) MERIS and OLCI images were used to derive chlorophyll-a
concentrations by using a SVR-based piecewise retrieval algorithm, and also detect algal bloom through
two indices. High temporal resolutions for MERIS (i.e., 3 days) and OLCI (i.e., 1-2 days) ensure to
provide sufficient observations on rapidly dynamic lake ecosystems. The averaged PEO values for
pixels within each lake were then obtained to delineate the eutrophication status and changes in fifty
large lakes of the Yangtze Plain during the MERIS (i.e., 2003-2011) and OLCI (i.e., 2017-2018)
observational periods. However, due to the unavailability of the crop- and nitrogen-related data for the
period of 2017-2018, we only used the PEO data derived from MERIS observations (i.e., 2003-2011)
here to examine their primary driving forces.

**2.5.2 Examination of the correlations between nutrient and PEO anomalies**

To examine the impacts of terrestrial nutrient sources on eutrophication changes in fifty large lakes of the Yangtze Plain, we used the simulated nitrogen leaching (LN) and anthropogenic phosphorus sources (i.e., total phosphorus from chemical fertilizer and manure, TP) representing the agricultural nutrient sources, and industrial wastewater discharge (IW) representing industrial nutrient sources. The gridded phosphorus fertilizer data were extracted from a global dataset developed by Lu and Tian (2017), while the phosphorus content in manure was calculated based on the nitrogen contents of manure products and the associated N:P ratios of different animals' excrement (Table S3). Annual industrial wastewater discharge data were obtained from the China City Statistical Yearbook (https://data.cnki.net/trade/Year-book/Single/N2018050234?zcode=Z011). Note that both agricultural phosphorus sources and industrial wastewater discharge are inventory data.

The 9-year mean (2003-2011) of three nutrient-related variables (i.e., LN, TP and IW) was used in a principal component analysis (PCA) followed by a K-means clustering (Hartigan and Wong, 1979) to classify examined fifty lakes based on similarities of terrestrial nutrient sources. In this process, all variables were normalized (across all years and lakes) based on the z-score method to remove the influence of different magnitudes in nutrient-related variables. We derived the first two principal components (PCs) from all normalized variables through a PCA, and the lakes were classified into three classes based on the first two PCs through the clustering methods. Finally, the annual anomalies of these nutrient-related variables and PEOs relative to their 9-year means were used to determine the primary drivers of temporal trends in eutrophication for each lake class.

**3 Results**

**3.1 Evaluation of LPJ-GUESS simulation**

For the evaluation of LPJ-GUESS simulation for the past four decades, the simulated LAI, GPP and crop yield were compared with observation-based estimates. Mean crop yields agreed well with the observed values, with mean relative errors of < 10% (Fig. 2). The comparison of simulated and observed LAI, and GPP were also satisfactory with overall high accuracy (i.e., a mean relative error of ~20% and

the root squared relative errors of < 30%) and spatial distributions consistent with observed patterns
(Fig. S1 and S2). Considering the difference in spatial scales between the grid cells and the gridded
evaluation data (i.e., the observed LAI and GPP maps), the overall performance of vegetation simulation
over the different land use types was considered acceptable. In addition, the simulated responses of
nitrogen leaching to different fertilizer applications at the experimental sites showed overall similar
trends as the observation ones for all three crops (i.e., maize, rice, and wheat), despite varying
magnitudes of differences between the simulated and observed leached nitrogen at certain fertilizer
level (Fig. 3).

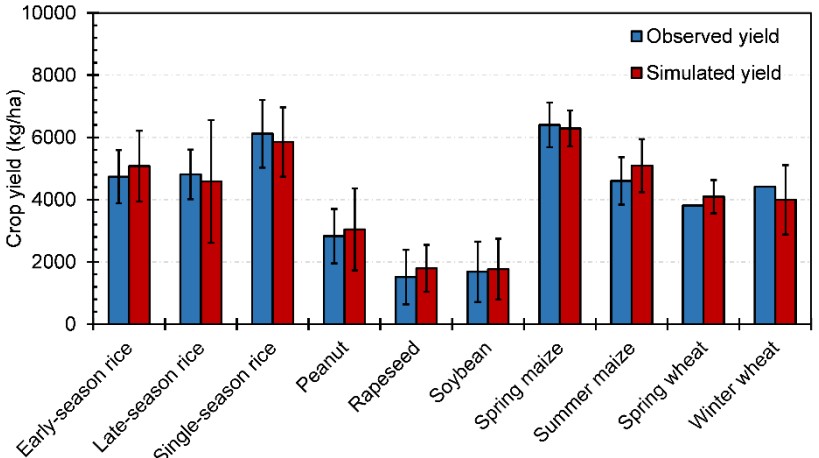


**Figure 2.** Comparison between the simulated and observed crop mean yields of different crops on the
Yangtze Plain; the mean values were averaged over the period 2000-2015 and across totally 179 sites.
Error bars show one standard deviation of crop yield.

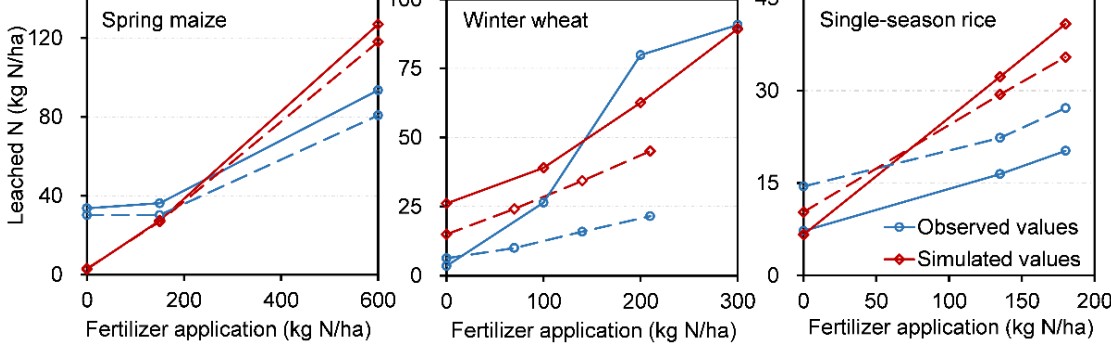


**Figure 3.** The simulated and observed responses of the leached nitrogen to different levels of fertilizer
application rates for three main crop types (i.e., maize, rice, and wheat) over the Yangtze Plain. Note
that the solid and dotted lines represented two different pairs of simulated and observed response of
leached nitrogen, respectively.

**3.2 Long-term changes of nitrogen use efficiency over the Yangtze Plain**

The average NUE for 1979 to 2018 was calculated to examine the spatial patterns of plant nitrogen
uptake on the Yangtze Plain. Considerable variations were detected across the entire Yangtze Plain,
with NUE values ranging from 5% to 60% (Fig. 4a). Two hotspots of high NUE were in the Hubei and
Jiangsu Province (see locations in Fig.1), dominated by cultivations of single-season rice and winter
wheat under the moderate levels (i.e., ~200 kg N ha$^{-1}$ yr$^{-1}$) of fertilizer applications (Fig. S3). The NUE
values also differed among different crop types for the past four decades. The largest NUE values were
found for soybean (74.0 % ± 11.0 %, Fig 5), while the lowest values were found for late-season rice

309    (15.9% ± 4.3%).

Due to the unprecedented increase of chemical fertilizer application since the 1980s, the crop NUE on
the Yangtze Plain has significantly decreased from ca. 50% in 1979 to 25% in 2018 ($p < 0.05$, in Fig.
4b), with an overall annual change rate of -0.55 % yr$^{-1}$. Overall, regions with relatively high levels of
NUE depicted a moderate or even slight increase for the past four decades, while the regions dominated
low-level NUE (i.e., Hubei and Hunan provinces in Fig. 1) experienced strongly declining trends (Fig.
4a&4c), as a result of the enhanced fertilizer applications. Considerable differences in magnitudes and
trends of NUE were also examined among the crop types. Significant decreases (t-test, $p < 0.05$) in the
decadal NUEs were found for seven crop types (annotated with "*" in Fig. 5), with the largest decrease
for the double cropping of early- and late-season rice (Fig. 4c and S3). In contrast, three crop types
experienced increasing trends of NUE, including peanut, spring wheat, and sugarcane (Fig. 5).

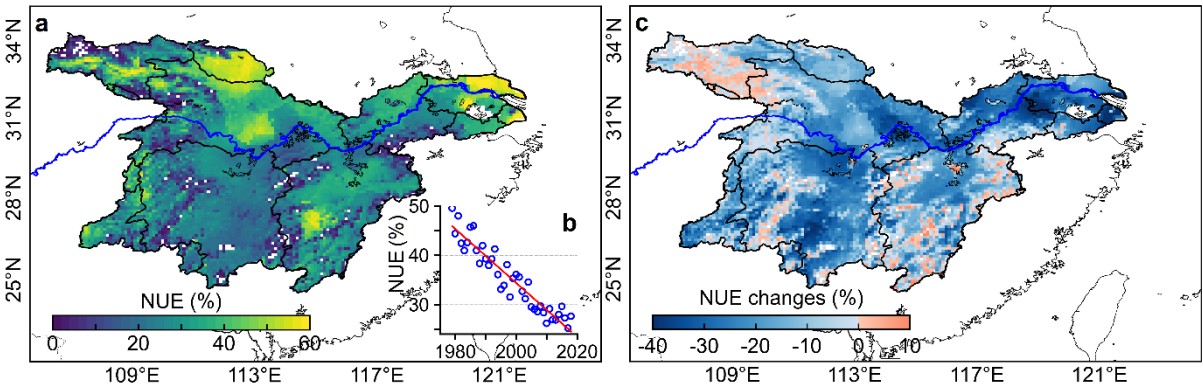

**Figure 4.** Nitrogen use efficiency (NUE) on the Yangtze Plain from 1979 to 2018. (**a**) Spatial distributions of climatological NUE (1979-2018), and the inset (**b**) shows the long-term trends of the area mean NUE; (**c**) Changes in NUE between the first (1979-1988) and the last (2009-2018) decades.

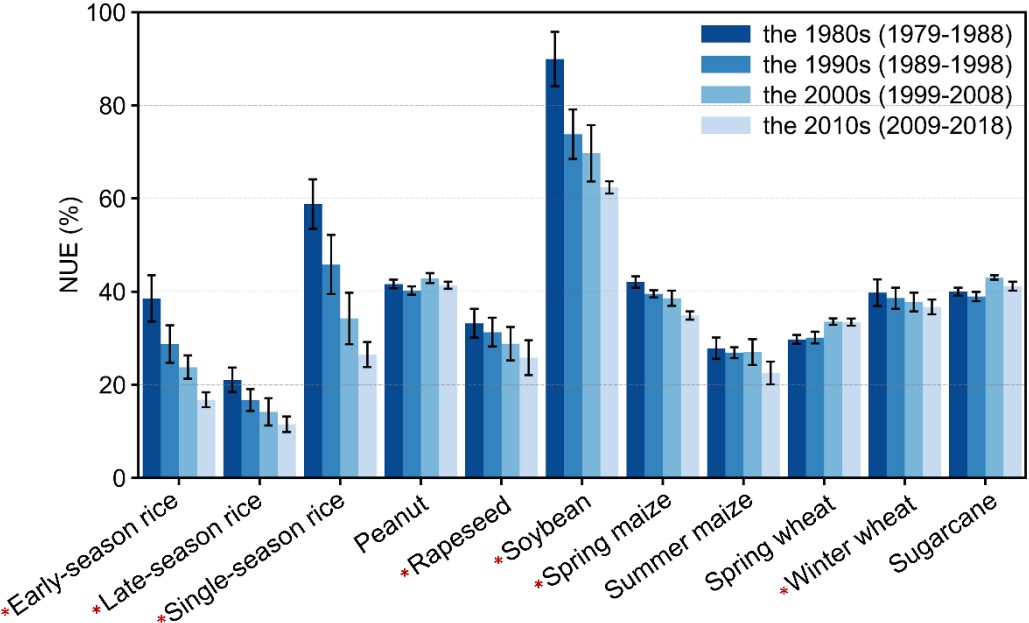

**Figure 5.** Decadal values of NUE for each crop functional type, averaged over the Yangtze Plain for the past four decades. Significantly decreasing trends ($p < 0.05$) are annotated with * using a t-test.

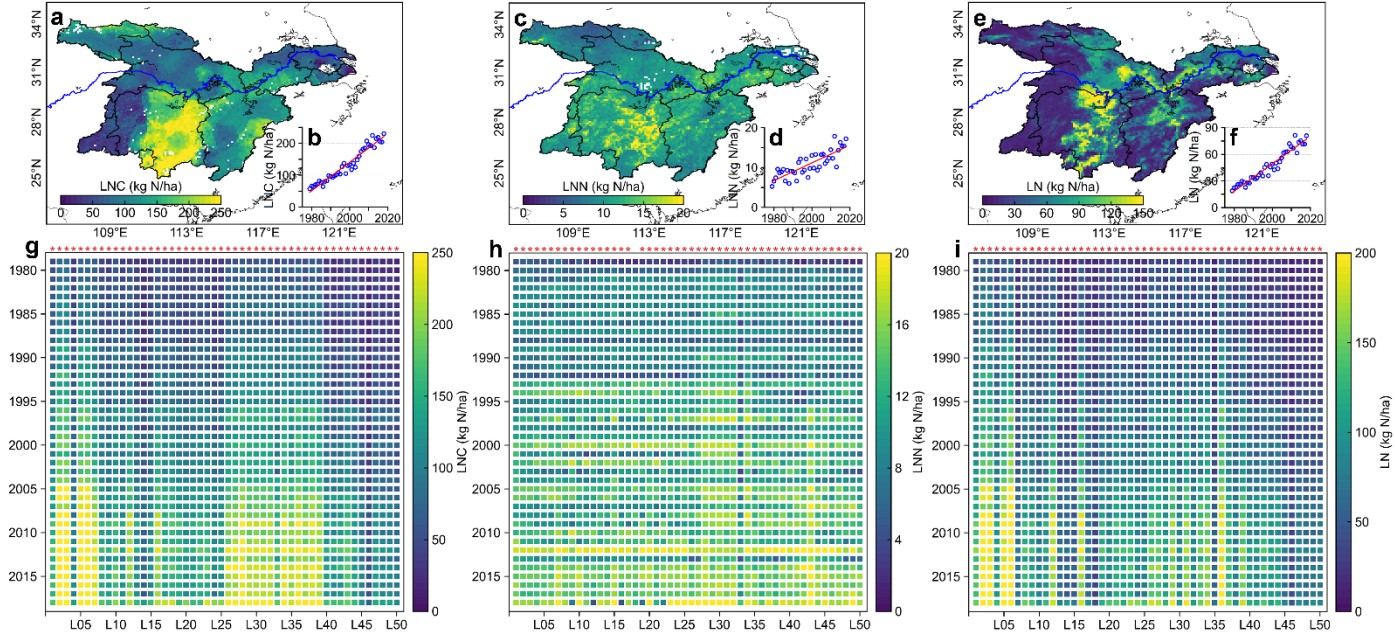

**Figure 6.** Tempo-spatial patterns of leached nitrogen from cropland (LNC), natural land (LNN), and total leached nitrogen (LN) for the period of 1979-2018. Spatial distributions of climatological (**a**) LNC, (**c**) LNN, and (**e**) LN. The insets (**b**), (**d**) and (**f**) represent the long-term changes of mean LNC, LNN and LN, where the red lines are the linear fitting lines between years and nitrogen leaching. Inter-annual changes of (**g**) LNC, (**h**) LNN, and (**i**) LN for all examined lakes (L01-L50) from 1979 to 2018. Statistically significantly positive trends ($p < 0.05$) are annotated with '*' on top of the panel, and see Fig. 1 for ID numbers of lakes.

## 3.3 Temporal and spatial patterns of nitrogen leaching for the past four decades

Along with the overall decreases in NUE, the leached nitrogen from both agricultural (LNC, averaged across cropland area) and natural systems (LNN, averaged across the natural area) experienced a statistically significant increase (t-test, $p < 0.05$) over the past four decades, with the different rates (4.5 kg N ha$^{-1}$yr$^{-2}$ and 0.22 kg N ha$^{-1}$ yr$^{-2}$ derived through the linear regression, respectively in Fig. 6b, 6d). The increased LNC was primarily associated with increased fertilizer applications (increased 2.5 times from 1979 to 2018), while the increased LNN was mainly linked to enhanced atmospheric deposition (explained 75.8% ± 6.8% of the increases in nitrogen sources) for natural ecosystems on the Yangtze Plain. The LNC were an order of magnitude larger than the LNN. The high levels of LNC were found

mainly in the Hunan Province (see Fig. 1 and Fig. 6a), with an average LNC value of 149 kg N ha$^{-1}$ yr$^{-}$
$^1$. In contrast, considerable spatial variations in LNN were revealed between the north and south parts
of the Yangtze Plain (Fig. 6c).
To understand nitrogen sources for each corresponding lake ecosystem on the Yangtze Plain, we
calculated the mean leached nitrogen (LN, averaged across the ground area) over the entire catchment
of each studied lake provided by the HydroLAKES dataset (Messager et al., 2016). The LN values
ranged from 29 kg N ha$^{-1}$ yr$^{-1}$ in Gehu Lake (L46 in Fig. 6i) to 153 kg N ha$^{-1}$ yr$^{-1}$ in Donghu Lake (L05
in Fig. 6i), indicating the considerable difference between the western lakes in the Hunan Province and
the eastern lakes in the Jiangsu Province. All examined lakes experienced statistically significantly
increasing trends in the LN (t-test, $p < 0.05$) over the past four decades (Fig. 6i), where the agricultural
activities contributed 94 % $\pm$ 5 % to the LN changes.
**3.4 Driving forces of terrestrial nutrient sources to eutrophication changes**
The leached nitrogen (LN), total phosphorus sources (TP), and industrial wastewater discharge (IW)
were used to represent terrestrial nutrient sources and were further investigated in terms of their linkages
to the observed PEOs. In the PCA analysis, the first two principal components (PCs) explained 48.7%
and 33.6% of variations in terrestrial nutrient sources (Fig. 7), where the first PC primarily depicts
positive dependence on IW but negative links with LN, and the second PC reveals negative dependences
on TP and IW. All fifty lakes were clustered into three classes based on the first two PCs (Fig. 7). Lakes
in class I (n = 22) had positive loading in the direction of the total phosphorus sources, with the main
coverage of the middle Yangtze Plain (i.e., Jiangxi and Anhui Province in Fig. 1), while class II cover
the most of lakes (n = 17) in the western regions (i.e., the Hunan Province and the western parts of the
Hubei Province in Fig. 1). The lakes of class III (n = 11) are primarily located on the eastern Yangtze
Plain, except for two lakes (i.e., Donghu and Tangxun lakes) which located at the urban area of Wuhan
City.
The correlations between annual anomalies of PEO and the three nutrient variables (relative to their
means for 2003-2011) were examined for all three lake classes. The PEO anomalies were significantly
correlated with different nutrient variables for three lake classes, indicating spatial variations of driving
factors for eutrophication changes on the Yangtze Plain (Fig. 8). Specifically, both LN and TP
anomalies exhibited significantly positive correlations ($p < 0.001$) with the PEO trends in lakes of class
I and II (Fig. 8a&b), indicating that the primary influence of agriculture-related sources to the increasing
trends of PEO. In contrast, the annual PEO dynamics in lakes of class III showed a significantly positive
correlation ($p < 0.05$) with industrial wastewater discharge (Fig. 8c), meaning that the temporal trends
of annual PEO in eastern parts of the Yangtze Plain were mainly associated with industrial wastewater
discharge. Note that the significantly negative correlations between the PEO and IW anomalies were
found for class I and II (Fig. 8c), which might be mechanistically unlikely. However, lakes in class I
and II are mainly located in western and central regions, with intensive agriculture activities and high
fertilizer applications (Chen et al., 2016). Such agriculture ecosystems provided substantial nutrient
sources for eutrophication growth and development, greatly larger than available nutrient from
industrial wastewater. In addition, agriculture nutrient sources generally increased with enhanced
fertilizer applications, while industrial wastewater discharge showed overall decreasing trends (Li et al.,
2013; Lyu et al., 2016). In such cases, industrial wastewater showed negative correlation with PEO
anomalies for class I and II lakes. It was also acknowledged that such correlation between industrial
wastewater and eutrophication changes might be affected by spatial variability in examined lakes within
each class.

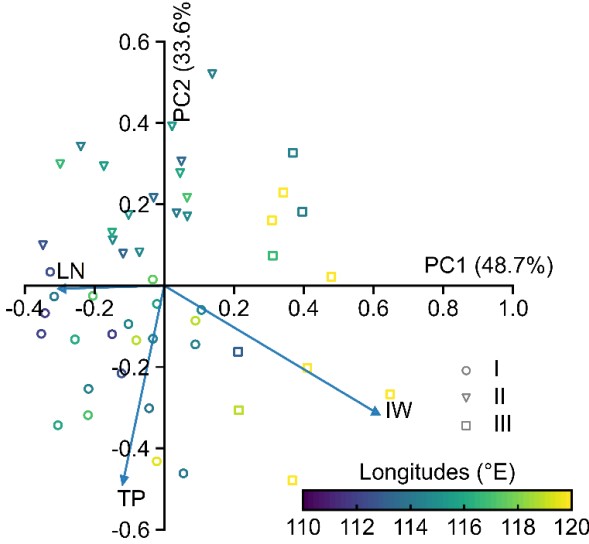


**Figure 7.** Loading plot of the principal component analysis (PCA) based on three nutrient-related variables. The color of scattering points represents the distributions of lakes in longitudinal order, and the directions of nutrient-related variables (i.e., LN leached nitrogen; TP total phosphorus sources; IW industrial wastewater) were annotated with blue arrows.

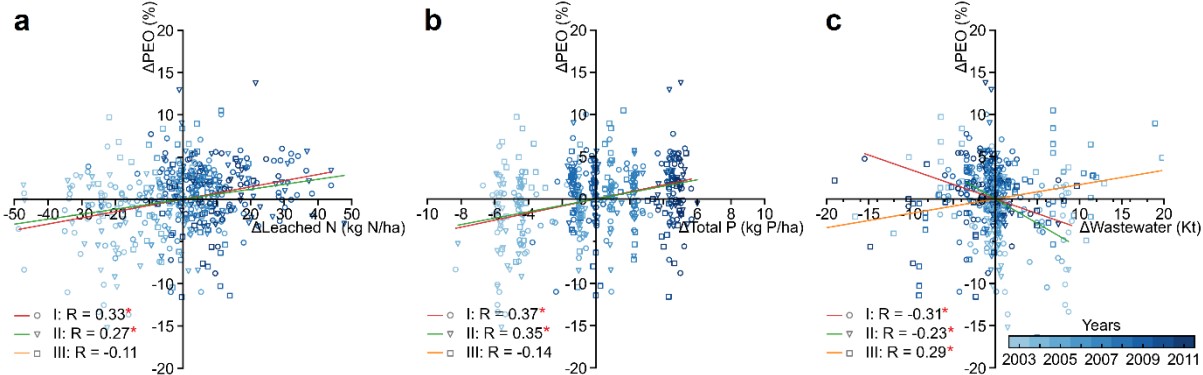

**Figure 8.** Relationships between the annual anomalies of PEO and nitrogen leaching **(a)**, total phosphorus sources **(b)**, and industrial wastewater discharge **(c)** for fifty studied lakes. The color and symbol of scattering points represent the years and lake classes, and the colored lines (shown for significant correlations only) are linear regressions between the annual anomalies of PEO and nutrient-related variables for each lake class. Significant correlation coefficients are marked by the red stars "∗".

## 4 Discussions

### 4.1 Significant decline in nitrogen use efficiency

The overall low mean NUE (27 %) and declining trends in NUE (-0.55 % yr$^{-1}$) characterized agricultural ecosystems on the Yangtze Plain for the past four decades, which are consistent with previous studies using statistical datasets and numerical modelling (Zhang et al., 2015; Yu et al., 2019). Over-fertilization was primarily responsible for decline in NUE from 1979 to 2018 (Shi et al., 2020; Zhang et al., 2015). Nitrogen fertilizer applications significantly increased by 2.5 times for past four decades, greatly exceeding the increase magnitudes in crop production (+26.3%), which potentially contributed to markedly decreasing NUE over the Yangtze Plain. Moreover, fertilization-induced increases of crop yield always decrease with the increase in fertilizer applications, and eventually disappear when crop yield reaches the upper limits (Zhang et al., 2015), suggesting that high fertilization rates are more likely

to generate the further decline in NUE over the Yangtze Plain. Over-fertilization might potentially
enhance nitrogen accumulation in soil that can be available for crop growth and development in next
years (Yang et al., 2006), thereby indicating that temporally increasing fertilization rates are generally
accompanied by declining NUE in agriculture ecosystems.
Considerable difference in NUE was examined among different crops, with the largest NUE values in
soybean for the past four decades (Fig. 5) as previously-documented NUE variations from 1961 to 2011
(Zhang et al., 2015). Generally, soybean has high NUEs mostly due to high protein contents (i.e., >
50%) in its grains (Fabre and Planchon, 2000). With the enhanced leaf nitrogen concentrations related
to its biological fixation, soybean tends to achieve a higher photosynthesis rate and delay leaf
senescence (Kaschuk et al., 2010; Ma et al., 2022), both of which potentially contributed to its generally
high NUE. Furthermore, double-cropping rice showed an overall lower NUE than single-season rice
(Fig. 5). It has been previously reported to occur in other double-cropping systems based on field
experiments, such as rice-wheat cropping (Liu et al., 2016b; Yi et al., 2015), rice-rapeseed cropping
(Wang et al., 2021a), and wheat-maize cropping (Xiao et al., 2021). Indeed, fertilizer applications
applied for the former crop could have accumulated nitrogen in soil that can be also taken by the latter
cultivated crop for their growth and development (Shi et al., 2020). In this regard, chemical fertilizer
applications for the latter crop can potentially generate the decline in its NUE.
**4.2 Primary causes of eutrophication changes.**
Our study revealed that the primary nutrient causes of eutrophication changes varied with regions over
the Yangtze Plain, where agricultural nutrient sources were strongly linked with eutrophication changes
in western and central lakes, while industrial wastewater showed a significantly positive correlation
with PEO trends in eastern lakes. Such spatial variations indicated that scientific policies and measures
were required to be implemented at local scales to mitigate eutrophication issues in lake ecosystems.
Separately, sustainable agriculture development should be encouraged to improve nitrogen/phosphorus
use efficiency and thus reduce agriculture nutrient sources available for western and central lakes to
potentially control eutrophication issues. In recent years, several agriculture practices have been
recommended and implemented, such as optimal fertilization schemes and residue removal, to pursue
high-efficiency agriculture on the Yangtze Plain (Cui et al., 2018; Shi et al., 2020). However,
smallholders were hesitant to adopt those knowledge-based practices, resulting in their poor
performance on agriculture sustainability (Cai et al., 2023). By contrast, national policies about
formulated fertilization was implemented in 2012, and fertilizer consumption started to decline since
2014 (Deng et al., 2021), which was expected to reduce agricultural nutrient sources in western and
central lakes.
In the eastern parts of the Yangtze Plain, policies and measures about mitigating eutrophication issues
were suggested to mainly focus on the decline and treatment in industrial sewage due to its large
contributions to nutrient exports delivered to lakes from the adjacent cities. The Jiangsu Province in the
eastern parts of the Yangtze Plain (see the locations in Fig. 1) experienced rapid economic and industrial
development since the policy of Reform and Opening-up of China since 1980s (Shen et al., 2020),
suggesting that the associated industrial wastewater discharge might be enhanced and then discharge
substantial nutrients to phytoplankton communities in lake ecosystems. In such cases, various national
strategies and policies have been gradually implemented to promote the green growth of industries on
the Yangtze Plain. Considerable efforts were made to encourage the reclamation of wastewater,
investment in the advances in wastewater treatment technology and installment of municipal wastewater
treatment plants (Li et al., 2013; Lyu et al., 2016). Furthermore, industrial structures were also
encouraged to transform from secondary to tertiary industries under the environment-friendly targets of
economic development (Huang et al., 2015). All these measures were expected to contribute to the
decline in industrial sewage on the Yangtze Plain.
**4.3 Limitations and Uncertainties**
Using the LPJ-GUESS model, we investigated the long-term changes and spatial variations of nitrogen
dynamics (i.e., plant nitrogen uptake and nitrogen leaching) over the Yangtze Plain for the past four
decades, and then examined the contributions of terrestrial nutrient sources to eutrophication changes
in fifty large lakes. However, due to the lacking representation of a phosphorus cycle in the LPJ-GUESS
model, we used external phosphorus fertilizer and manure application rates to represent the agricultural

phosphorus sources, without consideration of potential impacts from plant and soil processes. Phosphorus fertilizer applications significantly increased from 6.5 kg P ha$^{-1}$ in 1980 to 22.0 kg P ha$^{-1}$ in 2014, and previous studies also reported that the overall low phosphorus use efficiency (< 40%) characterized the Yangtze Plain from 2001 to 2015 (Zheng et al., 2018), both of which were similar to nitrogen patterns on the Yangtze Plain for the past four decades. In addition, the leached nitrogen showed strong dependence on fertilizer applications ($R^2$ = 0.92, $p$ < 0.001 in Fig. S4) over the Yangtze Plain for the past four decades. In this regard, we considered agricultural phosphorus sources as the potential driving force for eutrophication changes under the low levels of phosphorus use efficiency over the Yangtze Plain (Li et al., 2017; Zheng et al., 2018). Nevertheless, we also acknowledge that the use of phosphorus application data can generate uncertainties in our analysis, and thus processes related to phosphorus cycles are needed to add into LPJ-GUESS in the future to study the interactions of leached nitrogen and phosphorus on lake ecosystems.

Another source of uncertainty is associated with the transport processes that mediate the quantity and quality of terrestrial nutrients discharged to surface water ecosystems, as well as the impacts of aquaculture-related nutrient sources. Lateral transport rates of runoff and dissolved matter depend on soil properties, topography, and hydrological conditions over the drainage area (Solomon et al., 2015; Tang et al., 2014; Tang et al., 2018), which is required to further consider at regional scales to link to the dynamics of terrestrial nutrient exports for lake ecosystems on the Yangtze Plain. In addition, intensive and widespread freshwater aquaculture across the Yangtze Plain can contribute to accessible nutrient sources for eutrophication development and phytoplankton growth (Guo and Li, 2003; Wang et al., 2019a). Satellite observations revealed that 17 out of 50 lakes on the Yangtze Plain have established enclosure fishery nets to increase fish production (Dai et al., 2019). Consequently, substantial nutrients in fish food can directly enter aquaculture zones, promoting the contents of nitrogen and phosphorus in these lakes. These associated drivers are required to be comprehensively assessed to draw a complete picture of accessible nutrient sources for phytoplankton communities and then specify the anthropogenic impacts on water quality and eutrophication deterioration on the Yangtze Plain.

Uncertainties in the PEO data can originate from the uneven distributions of valid numbers of satellite
observations across the fifty large lakes of the Yangtze Plain. Under the influence of observational
conditions (i.e., cloud coverage and thick aerosols), the imagery with high-quality observations
distributed unevenly across the different years and seasons, which potentially resulted in certain impacts
on the derived annual PEOs and their temporal trends. Alternatively, the annual PEOs were calculated
based on the quarterly values to minimize such uncertainties. Nevertheless, more frequent satellite
observations (e.g., MODIS observations) will still be required to obtain a more accurate assessment of
eutrophication changes in lake ecosystems.

**5 Conclusions**

We used the LPJ-GUESS model to investigate the long-term changes of nitrogen dynamics over the
Yangtze Plain for the past four decades, and then examined their potential functions as the driving forces
of eutrophication changes in fifty large lakes of the Yangtze Plain. Significant decreases in NUE
dominated the whole Yangtze Plain, with the largest decrease in rice, soybean and rapeseed. The
leached nitrogen from both cropland and natural land showed statistically significant increasing trends
for all fifty examined lakes, indicating increased availability of terrestrial nitrogen sources in lake
systems for the past four decades. Two classes of lakes located in the western and central parts of the
Yangtze Plain showed significantly positive correlations between anomalies of PEO and agricultural
nutrient sources (i.e., the leached nitrogen and total phosphorus sources), and the PEO anomalies in the
remaining class (11 eastern lakes in the eastern parts of the Yangtze Plain) were positively correlated
with the industrial wastewater discharge. The impacts of agricultural and industrial nutrient sources on
eutrophication changes further emphasize the importance of region-specific policies and measures (i.e.,
sustainable management of agricultural nitrogen and phosphorus in western and central regions, and the
decline in wastewater-related nutrient discharge in eastern regions) to improve water environments.
*Code and Data availability*. The code of LPJ-GUESS model is stored in a central code repository and
will be made accessible upon request. Data used in this study are archived by the authors and are
available upon request.

*Author contributions*. QG, JT, LF and GS designed the framework and methodology of the study. QG

drafted the first version of the manuscript and analyzed the results. QG, JT and GS performed the

calibration of the LPJ-GUESS model. All co-authors contributed critically to the manuscript editing

and writing processes.

*Competing interests*. The authors declare that they have no conflict of interest.

*Acknowledgements*. This work was supported by the National Natural Science Foundation of China

(NOs: 41971304). Qi Guan was funded by the SUSTech-UCPH Joint Program. Jing Tang was

financially supported by Swedish FORMAS mobility grant (2016-01580) and MERGE Short project.

Stefan Olin acknowledges support from Lund University strong research areas MERGE and eSSENCE.

We are grateful to the European Space Agency (ESA) for publishing land cover dataset and to the China

Meteorological Data Service Center for providing crop distribution and yield data.

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
