# Peer review of "Long-term changes of nitrogen leaching and their contributions to lake"

_Biogeosciences, 2022_

## Author Comment (AC1)

**Response to reviewer 1:**

Using a process-based LPJ-GUESS model, this paper investigated the linkage between the dynamics of terrestrial nitrogen in the Yangtze Plain and eutrophication changes in fifty large lakes for the past four decades. Overall, the paper is well organized and the results answer the questions mentioned by the authors, although some uncertainties exist. But some details regarding the innovation and the method need to add before publication. The specific issues are as follows:

**Response:** Thank you for your constructive comments on improving our manuscript. Here, we have revised the manuscript according to these comments one by one.

**Specific comments:**

**1. Introduction**
Suggest to strengthen the innovation of the study.

**Response:** Thanks for this insightful suggestion. The innovation of this study includes 1) examining the regional scale's linkage between nutrient sources and eutrophication changes, and 2) explicit consideration of vegetation and soil nitrogen cycling as well as eco-hydrological processes on impacting terrestrial nutrient leaching. We have now added more descriptions about the innovation of this study as the text in lines 87-107: "To understand the primary causes of eutrophication in the lakes of the Yangtze Plain, previous studies have attempted to determine the contributions of riverine nutrient exports and lacustrine nutrient loading to algal blooms in individual lakes, such as Taihu and Chaohu lakes (Tong et al., 2017; Tong et al., 2021; Xu et al., 2015). Based on field-measured phytoplankton biomass and nutrient concentrations, algal blooms in Taihu Lake were primarily attributed to excessive nutrient loads from 1993 to 2015 (Zhang et al., 2018). Overloaded nutrients, in combination with climatic warming, were found to regulate the seasonal variations of cyanobacteria blooms in Chaohu Lake based on the monthly nutrient monitoring at discrete points (Tong et al., 2021). However, these studies only tracked the primary drivers of algal blooms for individual hyper-eutrophic lakes (i.e., Taihu and Chaohu lakes), which is insufficient to understand regional variations in terms of the causes of eutrophication and support the design of effective management strategies to mitigate eutrophication issues across different eutrophic states of lakes. Furthermore, lacustrine nutrient loading is always associated with terrestrial nutrient sources, such as synthetic fertilizers, livestock manure, and industrial sewage (Wang et al., 2019; Yu et al., 2018). For example, Wang et al. (2019) identified that diffuse sources contributed 90% to riverine exports of total dissolved nitrogen, and point sources discharged 52% of riverine phosphorus exports to Taihu Lake, where diffuse sources are synthetic fertilizers and atmospheric deposition, and point sources are sewage and manure discharge. It was also reported that chemical fertilizer and wastewater discharge provided primary nitrogen sources for the Chaohu Lake (Yu et al., 2018). Unfortunately, all these studies did not examine the impacts of vegetation uptake and soil retention on terrestrial nutrient sources, making it insufficient to comprehensively understand the linkage between terrestrial nutrient sources and eutrophication in regional lake ecosystems."

**2 Materials and Methods**
2.1 briefly introduce the agricultural cultivation types and management patterns in the study area.

**Response:** Thanks for this suggestion. Now, we have added a brief introduction about agricultural cultivation and management on the Yangtze Plain (lines 122-132): "Generally, rice-sown area

contributed dominantly to agriculture areas associated with climate conditions and human diet (Piao et al., 2010; Tilman et al., 2011). To enhance crop production, double-cropping strategy has been widely implemented on the Yangtze Plain, such as the rotation of early- and late-season rice (Chen et al., 2017), and the rotation of summer maize and winter wheat (Xiao et al., 2021). Several common management practices were adopted by millions of smallholders (Cui et al., 2018). For example, straw return, organic manure applications, and suitable planting density were also recommended in recent years (Cui et al., 2018). Significantly increased fertilizer applications to cropland were expected to stimulate crop yield over the past half century (Yu et al., 2019; Zhang et al., 2015). Such management practice can certainly enhance agriculture productivity, but also cause negative consequences to soil and aquatic environment (Liu et al., 2016; Shi et al., 2020)."

2.2 How the fertilization process is represented in the model needs to be described

**Response:** Thanks for this comment. We agree with you that the fertilization process deserves more detailed descriptions due to its importance to agricultural nitrogen leaching on the Yangtze Plain. The China Statistical Yearbook reported that chemical fertilizer applications increased from 55 kg N ha$^{-1}$ in 1980 to 137 kg N ha$^{-1}$ in 2014 to stimulate crop production over the Yangtze Plain (Chen et al., 2016). According to local farmers' practice (Shi et al., 2020), chemical fertilizer and livestock manure are often applied at three different stages: sowing, tillering, and heading stages for improving nitrogen use efficiency and nitrogen loss. Such fertilization schemes are represented in the LPJ-GUESS model (Olin et al., 2015), where nitrogen fertilizer is applied at crop development stages of 0, 0.5, and 0.9 which corresponds to the abovementioned three stages, and the relative fertilization rate for each stage are empirical parameters in LPJ-GUESS model. Now, these details about fertilization processes in the LPJ-GUESS simulation have been added to Section 2.2 (lines 162-167).

2.5.1 The remote sensing data of Guan, such as spatial and temporal resolution, should be briefly introduced.

**Response:** We agree with you that more details about remote sensing data used in Guan et al. (2020) should be added in the manuscript. We have added the following text on lines 249-252: "All full-resolution (300 m) MERIS and OLCI images were used to derive chlorophyll-a concentrations by using a SVR-based piecewise retrieval algorithm, and also detect algal bloom through two indices. High temporal resolutions for MERIS (i.e., 3 days) and OLCI (i.e., 1-2 days) ensure to provide sufficient observations on rapidly dynamic lake ecosystems."

**4 Discussions and conclusions**

(1) Lines 343-348 add a discussion of the reason for the decrease in NUE.

**Response:** Thanks for this comment. Now, we have added some discussion about the reasons for the decrease in NUE as lines 401-413: "The overall low mean NUE (27 %) and declining trends in NUE (-0.55 % yr$^{-1}$) characterized agricultural ecosystems on the Yangtze Plain for the past four decades, which are consistent with the previous studies using statistical datasets and numerical modelling (Zhang et al., 2015; Yu et al., 2019). Over-fertilization was primarily responsible for decline in NUE from 1979 to 2018 (Shi et al., 2020; Zhang et al., 2015). Nitrogen fertilizer applications significantly increased by 2.5 times for past four decades, greatly exceeding the increase magnitudes in crop production (+26.3%), which potentially contributed to markedly decreasing NUE over the Yangtze Plain. Moreover, fertilization-induced increases of crop yield always decrease with the increase in fertilizer applications,

and eventually disappear when crop yield reaches the upper limits (Zhang et al., 2015), suggesting that high fertilization rates are more likely to generate the further decline in NUE over the Yangtze Plain. Over-fertilization might potentially enhance nitrogen accumulation in soil that can be available for crop growth and development in next years (Yang et al., 2006), thereby indicating that temporally increasing fertilization rates are generally accompanied by declining NUE in agriculture ecosystems"

(2) add an outlook for future studies regarding the uncertainties of current results.

**Response:** Thanks for this suggestion. The uncertainties related to the current results include unrepresented phosphorus cycling in terrestrial ecosystems, the contributions of aquaculture-related nutrient sources, the regulation of transport processes to the amount of nutrient discharge to freshwater, and uneven distributions of satellite observations. We have some perspectives about future studies that can overcome these uncertainties as the text in lines 471-474: "Nevertheless, we also acknowledge that the use of phosphorus application data can generate uncertainties in our analysis, and thus processes related to phosphorus cycles are needed to add into LPJ-GUESS in the future to study the interactions of leached N&P on lake ecosystems", lines 477-480: "Lateral transport rates of runoff and dissolved matter depend on soil properties, topography, and hydrological conditions over the drainage area (Solomon et al., 2015; Tang et al., 2014; Tang et al., 2018), is required to further consider at regional scales to link to the dynamics of terrestrial nutrient exports for lake ecosystems on the Yangtze Plain", lines 486-489: "These associated drivers are required to draw a complete picture of accessible nutrient sources for phytoplankton communities and then specify the anthropogenic impacts on water quality and eutrophication deterioration on the Yangtze Plain.", and lines 495-497: "Nevertheless, more frequent satellite observations (e.g., MODIS observations) will still be required to obtain a more accurate assessment of eutrophication changes in lake ecosystems."

[revised manuscript text omitted]

---

## Author Comment (AC2)

**Response to reviewer 2:**

This study examined the dynamics of major pollution sources contributing to eutrophication in the fifty large lakes of the Yangtze River Plain, a region with 340 million people. These dynamics were studied over a forty-year period (1979-2018) and correlated with satellite-tracked algal bloom events. Pollutant sources studied include leached nitrogen in agro-ecosystems, phosphorus sources, and industrial waste. Remarkable efforts have been made to characterize in space and time the leaching of nitrogen which is a very diffuse source of pollution. This characterization was made using a carbon-nitrogen coupled ecosystem model constrained with meteorological, soil and agronomic data. This study is exceptional by the size of the studied region, the diversity of ecosystems considered (crops, terrestrial and aquatic ecosystems) and used variables. Below I made some comments/criticisms with the aim of improving this manuscript that merit publication.

**Response:** We sincerely thanks for your valuable comments and questions. We have addressed each in detail below.

The title and conclusive sentences of the abstract only mention the impact of N leaching. However, your study also suggests an impact of P applications and industrial wastes. This must appear more clearly.

**Response:** Thanks for spotting these out. We have changed the title of this manuscript to "Long-term changes of nitrogen leaching and the contributions of terrestrial nutrient sources to lake eutrophication dynamics on the Yangtze Plain, China" and also the last sentences of the abstract to: "Our results revealed the importance of terrestrial nutrient sources for long-term changes in eutrophic status over the fifty lakes of the Yangtze Plain. This calls for region-specific sustainable nutrient management (i.e., nitrogen and phosphorus applications in agriculture and industry) to improve the water quality of lake ecosystems."

From my point of view, on the most important finding of your study is the suggestion that the origins of lake eutrophication change with space (and maybe with time). Indeed, this finding signifies that policies and technical changes that need to be implemented to decrease lake eutrophication must be adapted to local conditions.

**Response:** Thanks for this comment. Our study concluded that the primary nutrient causes of eutrophication vary with space, suggesting that specific policies and measures are required to be implemented on local scales to improve eutrophication issues. Now, we have added some discussion into lines 428-432: "Our study revealed that the primary nutrient causes of eutrophication changes varied with regions over the Yangtze Plain, where agricultural nutrient sources were strongly linked with eutrophication changes in western and central lakes, while industrial wastewater showed a significantly positive correlation with PEO trends in eastern lakes. Such spatial variations indicated that scientific policies and measures were required to be implemented at local scales to mitigate eutrophication issues in lake ecosystems."

In the discussion, you mention that the eutrophication induced by industrial wastes is now solved thanks to policies set by the government and changes made in industries. YOU CAN NOT CONCLUDE on that point because your data do not show that at all. You can only mention the fact that the government has become aware of the problem and set up incentives to decrease industrial pollutions.

**Response:** Thanks for pointing out this issue. Yes, we agree with you that our analysis cannot reach the conclusions as eutrophication induced by industrial wastewater has been solved currently. Now, we have revised the discussion about the linkage between industrial wastewater and eutrophication changes in eastern lakes as lines 449-456: "In such cases, various national strategies and policies have been gradually implemented to promote the green growth of industries on the Yangtze Plain. Considerable efforts were made to encourage the reclamation of wastewater, investment in the advances in wastewater treatment technology and installment of municipal wastewater treatment plants (Li et al., 2013; Lyu et al., 2016). Furthermore, industrial structures were also encouraged to transform from secondary to tertiary industries under the environment-friendly targets of economic development (Huang et al., 2015). All these measures potentially contribute to the decline in industrial sewage on the Yangtze Plain."

The manuscript discussion starts by saying that it is now possible to deplete lake eutrophication because we know one of its main sources: low NUE and high N leaching. One might argues that we know since decades that N leaching causes eutrophication. The challenge rather lies in the setup of good agricultural practices leading to a decrease in nutrient loss. You should mention/list some examples of these practices (agriculture precision, cover cropping, residues managements etc) and how they are implemented or not in this region. More generally, I would suggest to start the discussion with your finding that the causes of eutrophication can be multiple and change in space requiring incentive policies adapted to local conditions.

**Response:** Thanks for this great suggestion and we have changed the order of the text in the discussion. We start with discussing spatial variations of potential causes of eutrophication on the Yangtze Plain, and what relevant policies and measures should be implemented at local scales to mitigate eutrophication issues (see lines 433-437). Separately, sustainable agriculture development should be encouraged to improve nitrogen/phosphorus use efficiency and thus reduce agriculture nutrient sources available for western and central lakes to potentially control eutrophication issues. In recent years, several agriculture practices have been recommended and implemented, such as optimal fertilization schemes and residue removal, to pursue high-efficiency agriculture on the Yangtze Plain (Cui et al., 2018; Shi et al., 2020). In the revised discussion, these potential management practices have been discussed in terms of their contributions to improving agriculture sustainability (see lines 435-442).

L153 It seems that soil properties have been implemented in the model using the world database on soil types/soil pedology. How did you consider the impact of land use on these soil properties?

**Response:** Not only soil type but also soil properties including fractions of sand, clay and silt, organic carbon content, C:N, pH, and bulk density, were used as LPJ-GUESS inputs. In the model, soil properties can influence soil hydrological and thermal properties and are not varying throughout the simulation period. For cropland areas, the model does consider the impacts of for instant, nutrients and crop residues on soil organic matter and nutrient contents, but does not simulate other land use impacts, such as grazing and soil compactions etc.

L199 This approach using the linear relationship is not clear? Why did you not directly use the model to estimate the N leaching?

**Response:** Thanks for pointing this out. We have directly used the modelled N leaching and the linear regression is just to obtain the changing rate/slope over time. Please see the revised text on lines 237-240: "We assessed long-term changes in nitrogen use efficiency (NUE) and nitrogen leaching over the past four decades. For the LPJ-GUESS simulated NUE and leached nitrogen, a linear regression was conducted on the annual mean values for the whole Yangtze Plain to determine the associated change rates (i.e., the regression slopes), and the significance was tested by a t-test."

L212-213 I do not understand why it is a problem since the model can provide those results.

**Response:** Thanks for this comment. We have used the same levels of fertilizer and manure applications in 2014 for the simulation years of 2015 to 2018 on the Yangtze Plain, due to the un-availability of gridded fertilizer and manure application data. This model setup might not represent the trends in later years, as there are some studies reporting the decreasing trends in total anthropogenic nitrogen input over the Yangtze Plain (Deng et al., 2021; Zhao et al., 2022). Furthermore, although the sown area of each crop changed slightly from year to year, the unavailability of crop cultivation information for the period of 2015-2018 would certainly introduce uncertainties to estimated crop nitrogen uptake. For all these above-mentioned reasons, we only selected the period when we have both PEO data and also the good quality of simulated leaching data, to examine their linkage with terrestrial nutrient sources on the Yangtze Plain.

L217 P sources are not simulated right? This should be specified.

**Response:** No, P sources were not simulated and they are inventory data. Now, it has been specified as the lines 267-268.

L225 These choices are a bit strange. In general, N (especially mineral N) is more mobile than P. Thus, N leaching is likely to be more linked to N applications than P leaching to P applications. Despite this, you used a complex modelling to estimate N leaching whereas you simply used the P source for P leaching/pollution. You should better explain your choices, and maybe consider to also examine the correlation between N leaching and N applications to have something comparable for the different sources of pollutions.

**Response:** We agree with the reviewer that N is more mobile than P. Our main argument is that previous studies have identified the low phosphorus use efficiency (< 40%) for agriculture ecosystems on the Yangtze Plain (Zheng et al., 2018). With the reported phosphorus fertilizer application increasing from 6.5 kg P ha$^{-1}$ in 1980 to 22.0 kg P ha$^{-1}$ in 2014, the large proportion of these added fertilizer will end up in the lake systems, suggesting that agricultural phosphorus sources can be considered as potential causes of eutrophication changes. In addition, the leached nitrogen also showed strong dependence ($R^2$ = 0.92, $p < 0.001$ in Fig. S4) on nitrogen applications over the Yangtze Plain for the past four decades. Now, we have added these details into the discussion (lines 464-469) about why we used agricultural phosphorus sources to examine the linkage between terrestrial nutrient sources and eutrophication changes on the Yangtze Plain.

L256 What do you mean with climatological NUE?

**Response:** The climatological NUE here represents the average NUE for the period of 1979 to 2018, and was used here to demonstrate spatial variations in NUE over the Yangtze Plain. Now, it has been clarified at lines 302-303: "The average NUE for 1979 to 2018 was calculated to examine the spatial patterns of plant nitrogen uptake on the Yangtze Plain".

L270-273 There are considerable differences between crops, which is worth to discuss. The negative effect of double crops is surprising since maintaining a permanent plant cover generally promote N retention in agrosystems.

**Response:** In terms of large difference in NUEs between crops, we have discussed their difference on lines 306-309: "The NUE values also differed among different crop types for the past four decades. The largest NUE values were found for soybean (74.0 % ± 11.0 %, Fig 5), while the lowest values were found for late-season rice (15.9% ± 4.3%)".

Now, we have added more discussion about comparison with previous studies, the reasons of large NUE difference among crops and the lower NUE for crop rotations as lines 414-426: "Considerable differences in NUE were examined among different crops, with the largest NUE values in soybean for the past four decades (Fig. 5) as previously-documented NUE variations from 1961 to 2011 (Zhang et al., 2015). Generally, soybean has high NUEs mostly due to high protein contents (i.e., > 50%) in its grains (Fabre and Planchon, 2000). With the enhanced leaf nitrogen concentrations related to its biological fixation, soybean tends to achieve a higher photosynthesis rate and delay leaf senescence (Kaschuk et al., 2010; Ma et al., 2022), both of which potentially contributed to its generally high NUE. Furthermore, double-cropping rice showed an overall lower NUE than single-season rice (Fig. 5). It has been previously reported to occur in other double-cropping systems based on field experiments, such as rice-wheat cropping (Liu et al., 2016; Yi et al., 2015), rice-rapeseed cropping (Wang et al., 2021), and wheat-maize cropping (Xiao et al., 2021). In this type of system, fertilizer applications applied for the former crop could have accumulated nitrogen in soil that can be also taken by the latter cultivated crop for their growth and development (Shi et al., 2020). In this regard, chemical fertilizer applications for the latter crop can potentially lead to lower NUE."

Figure 6 -> N leaching under natural ecosystems also increase during the period, which merits discussions.

**Response:** Thanks for this comment. Yes, the leached nitrogen from natural ecosystems also significantly increased for the past four decades (marked by red stars in Fig. 6h), which were primarily associated with increasing atmospheric deposition (Chen et al., 2016; Xu et al., 2018). Specifically, it was reported that nitrogen emission induced by industry development (i.e., chemical plants and factories) and livestock contributed dominantly to the growth of atmospheric deposition from 1980 to 2012 (Chen et al., 2016). Such enhanced atmospheric nitrogen deposition provided substantial nitrogen available for natural ecosystems and thus resulted in significantly increasing leached nitrogen. Now, all these descriptions have been added into lines 340-343.

L292-293 -> put the treatments close to the numbers to facilitate the reading.

**Response:** Thanks for this valuable suggestion. Now, I have put the treatments close to the numbers in line 339.

L298 How did you define the drainage area? Watershed?

**Response:** The drainage area of each lake indicates its catchment area, which is provided by the HydroLAKES dataset (Messager et al., 2016). Now, it has been revised as the entire catchment of each studied lake provided by the HydroLAKES dataset (line 349).

L303 What about the role of atmospheric N depositions? Did you consider this aspect or not? This should be specified.

**Response:** Yes, atmospheric nitrogen deposition is an input for the model (see description on lines 187-190), so contributes to the modelled nitrogen cycle. In our simulations, the atmospheric deposition is the main nitrogen sources (75.8% ± 6.8%) for natural ecosystems, while it only accounted for 5.8% ± 0.9% of nitrogen sources for agriculture ecosystems. Now, we have added the role of atmospheric N deposition in leached nitrogen from natural ecosystems into lines 340-343.

L305 -> please recall us the acronyms of these variable to facilitate the reading.

**Response:** We have added the acronyms of these three nutrient variables into the revised manuscript (line 356) to facilitate the reading.

L377 Revise the sentence. The source of uncertainty is not "potential impacts of terrestrial nutrient losses" but the transport processes that mediate the quantity and quality of nutrients that will be transferred from sources to surface water (lakes?)

**Response:** Thanks for this comment. This sentence has been revised as lines 475-476:" Another source of uncertainty is associated with the lateral transport processes that mediate the quantity and quality of terrestrial nutrients discharged to surface water ecosystems, as well as the impacts of aquaculture-related nutrient sources".

**Reference**

Chen, F., Hou, L., Liu, M., Zheng, Y., Yin, G., Lin, X., Li, X., Zong, H., Deng, F., and Gao, J.: Net anthropogenic nitrogen inputs (NANI) into the Yangtze River basin and the relationship with riverine nitrogen export, Journal of Geophysical Research: Biogeosciences, 121, 451-465, 2016.

Cui, Z., Zhang, H., Chen, X., Zhang, C., Ma, W., Huang, C., Zhang, W., Mi, G., Miao, Y., and Li, X.: Pursuing sustainable productivity with millions of smallholder farmers, Nature, 555, 363-366, 2018.

Deng, C., Liu, L., Peng, D., Li, H., Zhao, Z., Lyu, C., and Zhang, Z.: Net anthropogenic nitrogen and phosphorus inputs in the Yangtze River economic belt: spatiotemporal dynamics, attribution analysis, and diversity management, Journal of Hydrology, 597, 126221, 2021.

Fabre, F. and Planchon, C.: Nitrogen nutrition, yield and protein content in soybean, Plant Science, 152, 51-58, 2000.

Huang, C., Zhang, M., Zou, J., Zhu, A.-x., Chen, X., Mi, Y., Wang, Y., Yang, H., and Li, Y.: Changes in land use, climate and the environment during a period of rapid economic development in Jiangsu Province, China, Science of the Total Environment, 536, 173-181, 2015.

Kaschuk, G., Hungria, M., Leffelaar, P., Giller, K., and Kuyper, T.: Differences in photosynthetic behaviour and leaf senescence of soybean (Glycine max [L.] Merrill) dependent on N2 fixation or nitrate supply, Plant Biology, 12, 60-69, 2010.

Li, Y., Luo, X., Huang, X., Wang, D., and Zhang, W.: Life cycle assessment of a municipal wastewater treatment plant: a case study in Suzhou, China, Journal of cleaner production, 57, 221-227, 2013.

Liu, X., Xu, S., Zhang, J., Ding, Y., Li, G., Wang, S., Liu, Z., Tang, S., Ding, C., and Chen, L.: Effect of continuous reduction of nitrogen application to a rice-wheat rotation system in the middle-lower Yangtze River region (2013–2015), Field Crops Research, 196, 348-356, 2016.

Lyu, S., Chen, W., Zhang, W., Fan, Y., and Jiao, W.: Wastewater reclamation and reuse in China: opportunities and challenges, Journal of Environmental Sciences, 39, 86-96, 2016.

Ma, J., Olin, S., Anthoni, P., Rabin, S. S., Bayer, A. D., Nyawira, S. S., and Arneth, A.: Modeling symbiotic biological nitrogen fixation in grain legumes globally with LPJ-GUESS (v4. 0, r10285), Geoscientific Model Development, 15, 815-839, 2022.

Messager, M. L., Lehner, B., Grill, G., Nedeva, I., and Schmitt, O.: Estimating the volume and age of water stored in global lakes using a geo-statistical approach, Nature communications, 7, 13603, 2016.

Shi, X., Hu, K., Batchelor, W. D., Liang, H., Wu, Y., Wang, Q., Fu, J., Cui, X., and Zhou, F.: Exploring optimal nitrogen management strategies to mitigate nitrogen losses from paddy soil in the middle reaches of the Yangtze River, Agricultural Water Management, 228, 105877, 2020.

Wang, C., Yan, Z., Wang, Z., Batool, M., El-Badri, A. M., Bai, F., Li, Z., Wang, B., Zhou, G., and Kuai, J.: Subsoil tillage promotes root and shoot growth of rapeseed in paddy fields and dryland in Yangtze River Basin soils, European Journal of Agronomy, 130, 126351, 2021.

Xiao, Q., Dong, Z., Han, Y., Hu, L., Hu, D., and Zhu, B.: Impact of soil thickness on productivity and nitrate leaching from sloping cropland in the upper Yangtze River Basin, Agriculture, Ecosystems & Environment, 311, 107266, 2021.

Xu, W., Zhao, Y., Liu, X., Dore, A. J., Zhang, L., Liu, L., and Cheng, M.: Atmospheric nitrogen deposition in the Yangtze River basin: Spatial pattern and source attribution, Environmental Pollution, 232, 546-555, 2018.

Yi, Q., He, P., Zhang, X., Yang, L., and Xiong, G.: Optimizing fertilizer nitrogen for winter wheat production in Yangtze River region in China, Journal of Plant Nutrition, 38, 1639-1655, 2015.

Zhang, X., Davidson, E. A., Mauzerall, D. L., Searchinger, T. D., Dumas, P., and Shen, Y.: Managing nitrogen for sustainable development, Nature, 528, 51-59, 2015.

Zhao, Z., Zhang, L., and Deng, C.: Changes in net anthropogenic nitrogen and phosphorus inputs in the Yangtze River Economic Belt, China (1999–2018), Ecological Indicators, 145, 109674, 2022.

Zheng, J., Wang, W., Cao, X., Feng, X., Xing, W., Ding, Y., Dong, Q., and Shao, Q.: Responses of phosphorus use efficiency to human interference and climate change in the middle and lower reaches of the Yangtze River: historical simulation and future projections, Journal of Cleaner Production, 201, 403-415, 2018.

---

## Author Comment (AC3)

**Response to reviewer 3:**

This study focused on the dynamics of land-based nitrogen in 50 large lakes in the Yangtze River basin over the past 40 years, and used LPJ-GUESS model to study the driving factors of lake eutrophication. Based on the principal component analysis (PCA), the authors divided the 50 lakes into three types according to two principal components. This work is interesting because the authors have identified a driving mechanism for lake eutrophication. On the whole, I find it easy to understand and interesting. However, there are several issues in this manuscript that need to be corrected.

**Response:** Many thanks for your positive comments and we have revised my manuscript and addressed each comment below.

In line 31: It is suggested that the introduction should be supplemented with an explanation of the current state of eutrophication-driven research. There are some relevant studies in this area. Could you explain the innovation of your research?

**Response:** Thanks for this valuable suggestion. Now, we have briefly described the current state of eutrophication-driven research on the Yangtze Plain (see lines 66-86): "In recent several decades, national field surveys and satellite observations have been widely used to investigate nutrient loadings (Tong et al., 2017; Li et al., 2022), chlorophyll-a concentrations (Guan et al., 2020), trophic state index (TSI) (Hu et al., 2022; Chen et al., 2020) and algal bloom occurrence (Huang et al., 2020) to assess eutrophication issues in local or regional lakes of the Yangtze Plain, all of which revealed the lakes on the Yangtze Plain experienced eutrophication and algal blooms for the past two decades. Cyanobacteria blooms were reported to frequently occur in Taihu and Chaohu lakes, with the peak expanded extent reported for 2006 (Qin et al., 2019). Since then, the magnitudes of algal blooms significantly decreased from 2006 to 2013, and slightly increased again from 2013 to 2018 (Huang et al., 2020). Satellite observations revealed widespread and serious eutrophication issues in large lakes of the Yangtze Plain for the periods of 2003-2011 and 2017-2018, although significantly decreasing trends were found in 20 out of 50 lakes throughout the periods (Guan et al., 2020). Moreover, 35-year Landsat-derived trophic state index (TSI) also indicated that hyper-eutrophic and eutrophic lakes mainly characterized the Yangtze Plain, with slight increase in TSI from 1986 to 2012 and then decrease since 2012 (Hu et al., 2022). National field surveys demonstrated that although total phosphorus concentrations overall decreased from 2006 to 2014, it still remained under high levels (i.e., > 50 µg L$^{-1}$) in eastern China lakes (Tong et al., 2017). Various laws and guidelines were implemented on regional and national scales to control eutrophication problems, such as the Guidelines on Strengthening Water Environmental Protection for Critical Lakes in 2008 and the Water Pollution Control Action Plan in 2015 (Huang et al., 2019). Nevertheless, the eutrophication issues are still challenging to control and improve under the scarcity of effective strategies for the whole Yangtze Plain due to unknown causes of eutrophication issues."

The innovation of our study includes 1) examining the regional scale's linkage between nutrient sources and eutrophication changes, and 2) explicit consideration of vegetation and soil nitrogen cycling as well as eco-hydrological processes on impacting terrestrial nutrient leaching. Now, we have added more descriptions about the innovation of my research (lines 87-107): "To understand the primary causes of eutrophication in the lakes of the Yangtze Plain, previous studies have attempted to determine the contributions of riverine nutrient exports and lacustrine nutrient loading to algal blooms in individual

lakes, such as Taihu and Chaohu lakes (Tong et al., 2017; Tong et al., 2021; Xu et al., 2015). Based on field-measured phytoplankton biomass and nutrient concentrations, algal blooms in Taihu Lake were primarily attributed to excessive nutrient loads from 1993 to 2015 (Zhang et al., 2018). Overloaded nutrients, in combination with climatic warming, were found to regulate the seasonal variations of cyanobacteria blooms in Chaohu Lake based on the monthly nutrient monitoring at discrete points (Tong et al., 2021). However, these studies only tracked the primary drivers of algal blooms for individual hyper-eutrophic lakes (i.e., Taihu and Chaohu lakes), which is insufficient to understand spatial variations in causes of eutrophication changes on regional scales and support the design of region-specific effective management strategies to mitigate eutrophication issues. Furthermore, lacustrine nutrient loading is always associated with terrestrial nutrient sources, such synthetic fertilizers, livestock manure, and industrial sewage (Wang et al., 2019; Yu et al., 2018). For example, Wang et al. (2019) identified that diffuse sources contributed 90% to riverine exports of total dissolved nitrogen, and point sources discharged 52% of riverine phosphorus exports to Taihu Lake, where diffuse sources mainly focused on synthetic fertilizers and atmospheric deposition, while sewage and manure discharge represented point sources. It was also reported that chemical fertilizer and wastewater discharge provided primary nitrogen sources for Chaohu Lake (Yu et al., 2018). Unfortunately, all these studies did not examine the impacts of vegetation uptake and soil retention on terrestrial nutrient sources, making it insufficient to comprehensively understand the linkage between terrestrial nutrient sources and eutrophication in regional lake ecosystems."

In line 253: The simulated and observed responses of the nitrogen leaching to different levels of fertilizer application rates for three main crop types response in Figure 3. What do the dotted lines in Figure 3 mean? Please add the explanation.

**Response:** The dotted lines in Figure 3 are also the comparison between simulated (red) and observed (blue) response of leached nitrogen. In this study, we obtained two pairs of simulated and observed response of leached nitrogen for each main crop type, where two pairs were represented by the dotted and solid lines, respectively. Now, we have added more explanation into the figure caption.

In line 236: Observations of crop yields and GPP are used to assess the accuracy of the LPJ model in simulating the nitrogen leaching. Does this effectively evaluate the reliability of the model? Please explain the reason.

**Response:** Thanks for this comment. We think these two outputs (i.e., crop yield and GPP) are relevant for the assessed variable (i.e., leached nitrogen), although it could be more straightforward to compare with soil water nitrogen concentration or leaching if these data were available. As the model simulated crop developments in response to available nitrogen sources and environmental and soil conditions, the simulated crop yield and GPP can directly represent how well the model capture crop growth and vegetation dynamics, and indirectly evaluate the modelled nitrogen usage by plants. Therefore, we concluded that such evaluations can demonstrate the reliability of LPJ-GUESS simulations.

In line 274: In Figure 4 (b), check for missing units (%) in the vertical coordinates. Please check it.

**Response:** Yes, thanks for pointing out this issue. It definitely needs a units of NUE (%) that has been now added into Figure 4b.

In line 281: In Figure 6 (b) (d) and (f), the vertical coordinates lack units (Kg N/ha). Please check it.

**Response:** Thanks for this comment. All of these three figures should have units as kg N ha$^{-1}$ for the vertical coordinates. Now, I have added the units into these three figures.

In Figure S3, the position of the legend coincides with the picture. Please check and adjust the position of the legend.

**Response:** Thanks for pointing out this issue. The legend of Figure S3 has been moved to the right side, please see it in the supplementary file.

In line 324: Figure 7 (c) is not found in the manuscripts, and I assume you mean Figure 8 (c). Please check it.

**Response:** This is a typo. It should be Figure 8c that has been corrected now.

In line 335: In Figures 8 (a) and 8 (b), there are only relationships for type I and II lakes are found, but there no relationships for type III lakes. Please check it.

**Response:** Thanks for this comment. The leached nitrogen and agricultural phosphorus sources were found to positively correlate with the eutrophication changes for type I and II lakes (Fig. 8a&b). By contrast, the anomalies of industrial wastewater showed significantly positive correlations with the PEO trends in lake class III (R = 0.29, $p < 0.05$ in Fig. 8c). Now, we have used the red stars to mark statistically significant correlations in Fig. 8, and also added explanations into the figure caption.

In line 327: The significantly negative correlations between the PEO and IW Anomalies were found for Class I and II (Fig. 8C). This conclusion is very interesting and I hope the author can explain it.

**Response:** Thanks for this suggestion. Yes, the significantly negative correlations were found between industrial wastewater and PEO anomalies for Class I and Class II (Fig. 8c). As I mentioned above, lakes in Class I and II are mainly located in western and central regions, with intensive agriculture activities and high fertilizer applications (Chen et al., 2016). Such agriculture ecosystems provided substantial nutrient sources for eutrophication growth and development, which were greatly larger than available nutrient from industrial wastewater. In addition, agriculture nutrient sources generally increased with enhanced fertilizer applications, while industrial wastewater discharge showed overall decreasing trends (Li et al., 2013; Lyu et al., 2016). In such cases, industrial wastewater showed significantly negative correlation with PEO anomalies for Class I and II lakes. We also acknowledged that such correlation between industrial wastewater and eutrophication changes might be affected by spatial variability in examined lakes within each class. Now, we have added these explanations about the significantly negative correlations between the PEO and IW anomalies in class I and II lakes into Section 3.4 (see lines 378-387).

**Reference**

Chen, F., Hou, L., Liu, M., Zheng, Y., Yin, G., Lin, X., Li, X., Zong, H., Deng, F., and Gao, J.: Net anthropogenic nitrogen inputs (NANI) into the Yangtze River basin and the relationship with riverine nitrogen export, Journal of Geophysical Research: Biogeosciences, 121, 451-465, 2016.

Chen, Q., Huang, M., and Tang, X.: Eutrophication assessment of seasonal urban lakes in China Yangtze River Basin using Landsat 8-derived Forel-Ule index: A six-year (2013–2018) observation, Science of the Total Environment, 745, 135392, 2020.

Guan, Q., Feng, L., Hou, X., Schurgers, G., Zheng, Y., and Tang, J.: Eutrophication changes in fifty large lakes on the Yangtze Plain of China derived from MERIS and OLCI observations, Remote Sensing of Environment, 246, 111890, 2020.

Hu, M., Ma, R., Xiong, J., Wang, M., Cao, Z., and Xue, K.: Eutrophication state in the Eastern China based on Landsat 35-year observations, Remote Sensing of Environment, 277, 113057, 2022.

Huang, J., Zhang, Y., Arhonditsis, G. B., Gao, J., Chen, Q., and Peng, J.: The magnitude and drivers of harmful algal blooms in China's lakes and reservoirs: A national-scale characterization, Water Research, 181, 115902, 2020.

Huang, J., Zhang, Y., Arhonditsis, G. B., Gao, J., Chen, Q., Wu, N., Dong, F., and Shi, W.: How successful are the restoration efforts of China's lakes and reservoirs?, Environment international, 123, 96-103, 2019.

Li, S., Liu, C., Sun, P., and Ni, T.: Response of cyanobacterial bloom risk to nitrogen and phosphorus concentrations in large shallow lakes determined through geographical detector: A case study of Taihu Lake, China, Science of The Total Environment, 816, 151617, 2022.

Li, Y., Luo, X., Huang, X., Wang, D., and Zhang, W.: Life cycle assessment of a municipal wastewater treatment plant: a case study in Suzhou, China, Journal of cleaner production, 57, 221-227, 2013.

Lyu, S., Chen, W., Zhang, W., Fan, Y., and Jiao, W.: Wastewater reclamation and reuse in China: opportunities and challenges, Journal of Environmental Sciences, 39, 86-96, 2016.

Qin, B., Paerl, H. W., Brookes, J. D., Liu, J., Jeppesen, E., Zhu, G., Zhang, Y., Xu, H., Shi, K., and Deng, J.: Why Lake Taihu continues to be plagued with cyanobacterial blooms through 10 years (2007–2017) efforts, Science Bulletin, 64, 2019.

Tong, Y., Xiwen, X., Miao, Q., Jingjing, S., Yiyan, Z., Wei, Z., Mengzhu, W., Xuejun, W., and Yang, Z.: Lake warming intensifies the seasonal pattern of internal nutrient cycling in the eutrophic lake and potential impacts on algal blooms, Water Research, 188, 116570, 2021.

Tong, Y., Zhang, W., Wang, X., Couture, R.-M., Larssen, T., Zhao, Y., Li, J., Liang, H., Liu, X., and Bu, X.: Decline in Chinese lake phosphorus concentration accompanied by shift in sources since 2006, Nature Geoscience, 10, 507-511, 2017.

Wang, M., Strokal, M., Burek, P., Kroeze, C., Ma, L., and Janssen, A. B.: Excess nutrient loads to Lake Taihu: Opportunities for nutrient reduction, Science of the Total Environment, 664, 865-873, 2019.

Xu, H., Paerl, H., Qin, B., Zhu, G., Hall, N., and Wu, Y.: Determining critical nutrient thresholds needed to control harmful cyanobacterial blooms in eutrophic Lake Taihu, China, Environmental science & technology, 49, 1051-1059, 2015.

Yu, Q., Wang, F., Li, X., Yan, W., Li, Y., and Lv, S.: Tracking nitrate sources in the Chaohu Lake, China, using the nitrogen and oxygen isotopic approach, Environmental Science and Pollution Research, 25, 19518-19529, 2018.

Zhang, M., Shi, X., Yang, Z., Yu, Y., Shi, L., and Qin, B.: Long-term dynamics and drivers of phytoplankton biomass in eutrophic Lake Taihu, Science of the Total Environment, 645, 876-886, 2018.